# Prevalence and Demographic, Socioeconomic, and Behavioral Risk Factors of Self-Reported Symptoms of Sexually Transmitted Infections (STIs) among Ever-Married Women: Evidence from Nationally Representative Surveys in Bangladesh

**DOI:** 10.3390/ijerph19031906

**Published:** 2022-02-08

**Authors:** Md. Nazmul Huda, Moin Uddin Ahmed, Md. Bakhtiar Uddin, Md Kamrul Hasan, Jalal Uddin, Tinashe Moira Dune

**Affiliations:** 1School of Population Health, University of New South Wales, Kensington, NSW 2052, Australia; hudasoc2020@gmail.com; 2School of Health Sciences, Western Sydney University, Campbelltown, NSW 2560, Australia; 3School of Liberal Arts and Social Sciences, Independent University, Dhaka 1229, Bangladesh; 4Translation Health Research Institute, Western Sydney University, Campbelltown, NSW 2560, Australia; moin.ahmed@westernsydney.edu.au; 5Institute of Health Economics, University of Dhaka, Dhaka 1000, Bangladesh; 6School of Social Sciences, University of New South Wales, Kensington, NSW 2052, Australia; bakhtiareco@gmail.com; 7Department of Economics, Jatiya Kabi Kazi Nazrul Islam University, Mymensingh 2220, Bangladesh; 8Bangladesh Institute of Social Research Trust, Dhaka 1207, Bangladesh; mhas089@aucklanduni.ac.nz; 9Department of Community Health and Epidemiology, Dalhousie University, Halifax, NS B3H 1V7, Canada; jalal.uddin@dal.ca; 10Diabetes, Metabolism and Obesity Translational Research Unit, Western Sydney University, Campbelltown, NSW 2560, Australia

**Keywords:** sexually transmitted infection, abnormal genital discharge, genital sores, determinants, ever-married women, sexual and reproductive health, Bangladesh

## Abstract

Sexually transmitted infections (STI) symptoms (e.g., abnormal genital discharge and genital sores/ulcers) are a major public health concern in Bangladesh because the symptoms can indicate an STI and cause sexual and reproductive health complications in women of reproductive age. To our knowledge, no study examined the prevalence and risk factors of STI symptoms using a nationally representative sample. This study investigates the prevalence of STI symptoms among ever-married women in Bangladesh and the associations of STI symptoms with various demographic, socioeconomic, and behavioral risk factors using the most recent available data (2007, 2011, and 2014) of the Bangladesh Demographic and Health Surveys (BDHS). The BDHS employs a two-stage stratified sampling technique. The analytic sample comprised 41,777 women of reproductive age (15–49 years). Outcome variables included STI symptoms: abnormal genital discharge and genital sores/ulcers. Multivariate logistic regression was employed to find the adjusted odds ratio with a 95% confidence interval to assess the associations of outcome measures with explanatory variables. The study found that the prevalence of abnormal genital discharge and genital sores/ulcers among ever-married women aged 15–49 years was 10% and 6%, respectively. Multivariable analysis revealed that for women aged 25–34 years, those who used contraceptives and married earlier had an increased likelihood of STI symptoms. Furthermore, women from the wealthiest wealth quintile and couple’s joint decision-making were less likely to have STI symptoms. Findings have implications for interventions efforts aiming to improve women’s sexual and reproductive health in Bangladesh.

## 1. Introduction

Sexually transmitted infections (STIs) (e.g., gonorrhea, chlamydia, syphilis, and trichomonas) are generally transmitted through sexual contact and can have some symptoms (including abnormal genital discharge, genital sores/ulcers), representing a severe public health concern [1]. The STI symptoms can cause sexual and reproductive health (SRH) complications in women of reproductive age [1], infertility, and mother-to-child transmission if left untreated and undetected [2,3,4,5]. Furthermore, STIs may heighten individuals’ risk for acquiring or transmitting HIV three times or more [2]. The prevalence of STIs is increasing globally [1]. Daily, more than 1 million men and women aged between 15 and 49 years manifest STI symptoms [2]. Heightened biological risk for contracting STIs [6], inadequate access to sexual health services [7], and limited agency to negotiate condom use with their partners [8] put women at increased risk for STIs [9]. The highest 40% of global STI burdens occurred in women in Sub-Saharan Africa, followed by the Southeast Asia region (20%) [10]. In Bangladesh, various studies report an ever-increasing rate of STI among women of reproductive age, ranging from 0.2% to 39.1% [11,12,13,14]. The increasing STIs burden remains a major public health concern in Bangladesh, especially among women.

Existing studies show mixed relationships of various demographic, socioeconomic, and behavioral risk factors with STIs, mostly with the clinical outcomes, such as chlamydia, gonorrhea, genital herpes, syphilis, and trichomoniasis [15,16,17,18,19,20,21]. For example, some population-level studies from low-income countries find that women with lower social status (e.g., lower levels of education, lower income, etc.) have an increased risk for STIs [5,20,22]. Another study reports that women in the wealthiest quintile had a lower risk of STIs [21]. Studies have also shown that women who encountered spousal abuse were significantly more likely to report STIs [17,18,19]. Younger age [15,16], being married [23], and contraceptives use [18] were associated with increased STIs. Further, women who solely took decisions about own healthcare were less likely to report STI [17]. Thus, existing literature suggests that women’s STIs are linked to various domains of risk factors. 

Although a growing body of global studies investigates the associations of STIs with diverse risk factors, fewer studies focus on multiple domains of STI risk factors in a single study using nationally representative samples. In particular, Bangladeshi studies have been limited by the use of nonrepresentative samples, specific occupational groups and subpopulations. For instance, some earlier cross-sectional studies examined only sociodemographic factors (e.g., age, gender, marital status, and education) among men and women [24]. Others examined behavioral factors such as condom use, frequency of intercourse, and the number of sex partners [12,14]. Most of these studies were conducted among the specific subpopulations, such as adolescents [24], truck stand workers [14,25], and hospital patients [12]. These studies generally suffer from limited generalizability due to their focus on subgroups with nonrepresentative samples. Existing literature also tended to overlook the role of multiple domains of risk factors and how they complexly predict STI symptoms. To the best of our knowledge, in Bangladesh, no recent study examines the associations of women’s self-reported STI symptoms with multiple domains of risk factors using a nationally representative sample. Using data from three rounds of a nationally representative population-based survey, we address this research gap by examining the associations of demographic, socioeconomic, and behavioral risk factors with self-reported STI symptoms among ever-married women in Bangladesh. The country is marked by conservative sociocultural values and a patriarchal family structure in which men are considered the breadwinners and heads of households. Such socioeconomic contexts may benefit men and put women in a disadvantaged position, thus limiting their agency [9]. The current study provides a scientific understanding of the self-reported prevalence of STI symptoms and the demographic, socioeconomic, and behavioral risk factors of STI symptoms, which are crucial to inform public health interventions in Bangladesh from a policy perspective.

## 2. Materials and Methods

### 2.1. Data Sources and Sample

This study used data from the three most recent rounds (2007, 2011, and 2014) of the Bangladesh Demographic and Health Survey (BDHS). The latest 2017–18 BDHS did not collect data on the STI symptoms. The BDHS is a nationally representative cross-sectional household survey conducted every three years as a part of the global Monitoring and Evaluation to Assess and Use Results Demographic and Health Surveys (MEASURE DHS) programs (https://dhsprogram.com/ accessed on 30 January 2022). Worldwide, MEASURE DHS Programs are widely acknowledged and established data sources. The DHS Programs provide technical assistance to execute more than 320 household and facility-based surveys in 90 countries, including Bangladesh [26]. The BDHS applied a two-stage stratified sampling technique to collect data on various demographic, socioeconomic, and behavioral risk factors and STI symptoms from ever-married women. Study participants were ever-married women of reproductive age (15–49 years) from Bangladesh. Details about the data collection and sampling procedure can be found elsewhere [26].

### 2.2. Outcome Variables

Outcome measures considered in this study were self-reported STI symptoms such as abnormal genital discharge and genital sores/ulcers. Following previous studies [12,18,19,22,27,28], two binary variables were constructed with 1 if the participants responded “yes” to the questions: “Have you had abnormal genital discharge in the last 12 months preceding the survey?” and “Have you had genital sores/ulcers in the last 12 months preceding the survey?”, and 0 otherwise. Therefore, self-reported STI symptoms were defined as having abnormal genital discharge and genital sores/ulcers in the last 12 months preceding the survey.

### 2.3. Explanatory Variables

Explanatory variables were categorized into demographic, socioeconomic, and behavioral factors (Table 1). The choice of these variables and categorization were based on the extensive review of existing Bangladeshi and international studies [18,22,24,27,28,29,30,31,32]. Demographic factors included in the study were participants’ age (15–24, 25–34, 35–49 years), age at first marriage (9–14 years, 15–17 years, ≥18 years), and residence type (urban, rural). Socioeconomic factors included participants’ education level (no education, primary, secondary, higher), partners’ level of education (no education, primary, secondary, higher), wealth quintile (poorest, poorer, middle, richer, richest), and participants’ paid work status (no, yes). The behavioral factors were the current use of contraceptive methods (no method, traditional method, modern method), knowledge about STIs (no, yes), wife-beating justified if she refuses to have sex (no, yes), women’s healthcare decision-making (wife, husband, joint decisions, respondent, and someone else), and mass media exposure (not at all, irregular, regular).

### 2.4. Statistical Analysis

The analysis was based on a pooled sample of three rounds of BDHS (2007, 2011, and 2014). Our analytic sample yielded 41,777 ever-married women aged 15–49 years after dealing with missing cases. The prevalence of two outcome measures of STI symptoms was estimated by key risk factors. We presented the study sample’s characteristics by two outcomes of interest. The significance of difference was assessed with Pearson’s chi-square test. We evaluated the associations of STI outcomes with demographic, socioeconomic, and behavioral risk factors using three separate logistic regression models containing variables with a *p*-value below 0.20 from the earlier set of chi-square tests [33]. The first model adjusted for age, age at first marriage, and place of residence. The second model further adjusted for socioeconomic factors, such as women and their partner’s education, wealth index, and paid work status. The third model additionally included contraceptive use, STI knowledge, whether wife-beating is justified if she refuses to have sex, exposure to mass media, women’s healthcare decision-making, and survey year. Analyses were performed using the statistical software Stata (version 16.1). To adjust for population differences in each survey, we denormalized sampling weights following the DHS Sampling and Household Listing Manual [34]. A sensitivity analysis with an imputed dataset assessed the possible influence of missing data on the calculated coefficients [35,36]. 

## 3. Results

### 3.1. Sample Characteristics

The characteristics of the sample participants are presented in Table 2 in terms of abnormal genital discharge and genital sores/ulcers. More than one-third of the participants aged 25–34 years had abnormal genital discharge or genital sores/ulcers. Most of the respondents with STI symptoms (about 44%) had their first marriage when they were 9–14 years old. Three out of four participants reported living in rural areas. Table 2 also shows that approximately 26% of the women with abnormal genital discharge or genital sores/ulcers had no education, whereas around 30% of women’s partners had no formal education. More than one-fifth of the women with STI symptoms were from the richest wealth quintile, and 79% had no paid work. Nearly two-fifths of women with STI symptoms did not use any method of contraception. Approximately three-fourths of women had STI knowledge. Most women (92%) responded that beating by husbands was not justified if women refused to have sex with them. Around half of the women made joint decisions with their husbands regarding their healthcare, and 35% of women had no exposure to mass media.

### 3.2. Prevalence of STI Symptoms by Key Risk Factors

Figure 1 presents the prevalence of abnormal genital discharge and genital sores/ulcers by key risk factors. Our analyses found that the overall prevalence of abnormal genital discharge and genital sores was 10% and 6%, respectively (indicated by dashed lines in Figure 1), during 2007–2014 among women aged 15–49 years in Bangladesh. Findings also indicate that STI symptoms were most prevalent among women aged 25–34 years, first married at 9–14 years, lived in rural areas, had primary level education, and from poorer households. In addition, women who used traditional and modern contraceptive methods reported having a higher rate of abnormal genital discharge and genital sores.

### 3.3. Associations of STI Symptoms with Demographic, Socioeconomic, and Behavioral Risk Factors

Table 3 presents the adjusted odds ratio (AOR) with a 95% confidence interval (CI) obtained from the multivariable logistic regression on the association of genital discharge with demographic, socioeconomic, and behavioral risk factors. We showed results from models 1 and 2 to compare changes in coefficients in model 3. While our final model (model 3) included all study factors, models 1 and 2 included demographic and socioeconomic factors. The final model revealed that women aged 25–34 had 14% higher odds of having abnormal genital discharge (AOR: 1.14; 95% CI: 1.03, 1.27) compared with women aged 15–24. The odds of experiencing abnormal genital discharge significantly increased by 21% (AOR: 1.21; 95% CI: 1.08, 1.36) for women who married first at 9–14 years, compared to women who married at 18 years or older. Women who completed higher education had significantly lower odds of having abnormal genital discharge (AOR: 0.79; CI: 0.63, 1.00) than women with no education. Women in the richest wealth quintile had a 26% lower likelihood of (AOR: 0.74; 95% CI: 0.62–0.88) experiencing abnormal genital discharge than those in the lowest wealth quintile. Women with paid work were 13% (AOR: 1.13; 95% CI: 1.02, 1.26) more likely to have abnormal genital discharge.

Model 3 in Table 3 also shows that women who used traditional method and modern contraception methods were 42% (AOR: 1.42; 95% CI: 1.23, 1.64) and 9% (AOR: 1.09; 95% CI: 1.00, 1.19) more likely to experience abnormal genital discharge, respectively, compared to women who did not use any contraceptive methods. Furthermore, women who made joint decisions with their husbands regarding healthcare were less likely (AOR: 0.76, 95% CI: 0.68, 0.85) to experience abnormal genital discharge, compared to women who solely made such decisions. Women who had irregular (AOR: 1.22; 95% CI: 1.07–1.38) and regular (AOR: 1.11; 95% CI: 1.00–1.23) exposure to mass media had higher odds of having abnormal genital discharge than those who had no exposure at all. 

Table 4 shows the associations of genital sores/ulcers with demographic, socioeconomic, and behavioral factors. In model 3, women aged 25–34 years had a higher likelihood of genital sores/ulcers (AOR: 1.21, 95% CI: 1.06, 1.37) compared to women aged 15–24. Women in the richest wealth quintile had 25% lower odds of having genital sores/ulcers (AOR: 0.75, 95% CI: 0.61, 0.93) than those in the poorest wealth quintile. Women involved in paid work had higher odds of (AOR: 1.20; 95% CI: 1.07, 1.36) abnormal genital discharge than those who were not involved in a paid work. Women with STI knowledge had higher genital sores/ulcers (AOR: 1.26, 95% CI: 1.11, 1.42). Women who supported that wife-beating was justified if they refused to have sex had higher genital sores/ulcers (AOR: 1.27, 95% CI: 1.08, 1.49) than those who thought it was not justified. Regarding own healthcare, women who made joint decisions with their husbands (AOR: 0.81, 95% CI: 0.70, 0.93) and those who made decisions with someone else (AOR: 0.78, 95% CI: 0.62, 0.98) had lower odds of genital sores/ulcers, compared to women who independently made decisions regarding their healthcare. The revised AORs from the multiple imputation analysis did not substantially vary from the complete case analysis (Appendix A), indicating that missing data did not significantly alter the observed results.

## 4. Discussion

This study examined the associations of demographic, socioeconomic, and behavioral risk factors with STI symptoms among ever-married women of reproductive age in Bangladesh. The analyses indicate that some subgroups of women, for instance, women aged 25–34 and who had paid work, were more likely to experience STI symptoms. Women from the wealthiest households and those who made joint decisions regarding their healthcare were advantageous in reporting lower STI symptoms. Additionally, women who used contraceptive methods and had exposure to mass media tended to have higher abnormal genital discharge than those who used no contraceptive methods and had no exposure to mass media. The observed associations of STI symptoms with demographic, socioeconomic, and behavioral risk factors moderately concur with previous studies. The studies demonstrated increased STI symptoms among women aged 25–34 years [16,24], those with limited decision-making capacity regarding healthcare [17], those with lower levels of education [22], and women who used contraceptives such as condoms and oral pills [18]. Thus, the current study’s findings contribute to a growing body of literature [9,16,18,19,22,24,27,28] that has documented the association of STI symptoms with varied demographic, socioeconomic, and behavioral risk factors. 

This analysis identified age as a key demographic risk factor for STI symptoms. Women aged 25–34 consistently reported more STI symptoms than younger women (15–24 years). This finding aligns with existing literature which suggests that participants aged 25–34 years and older were more likely to have genital symptoms [16]. Women at this specific childbearing age (25–34 years) may have a higher coital frequency than younger cohorts (15–24 years) and are more likely to have unprotected sex with their husbands to meet the demands of having children, especially in low-income settings [18] such as Bangladesh. Such unprotected intercourse may increase the likelihood of STI symptoms among childbearing women [37]. This finding of our study may also suggest that STI awareness programs may not have protected young women from contracting STI symptoms in Bangladesh. Therefore, policy makers should promote age-specific interventions for improving young women’s sexual health.

Consistent with previous studies [18,38], our analyses found that child marriage (9–14 years) is associated with increased odds of abnormal genital discharge. This finding indicates that very early marriage may reduce women’s agency to negotiate safe intercourse with their partners and increase the likelihood of abnormal genital discharge. It is well established that women who are married early are financially dependent on their husbands, have limited education, health literacy, and employment [37,39]. Women also have inadequate knowledge about sexual and reproductive health issues, limited decision-making power regarding their own healthcare issues, and less access to healthcare facilities [39,40]. Together with the pervasive patriarchal family relations in Bangladesh, these factors may make younger women, compared to their older counterparts, more vulnerable to unprotected intercourse and expose them to STI risks [37]. In this regard, intervention programs need to give more attention to women who marry early. Therefore, intervention programs need to promote girls’ education that often helps delay marriage to reduce the risk of STI infections. Currently, due to the COVID-19 pandemic, there has been an increase in child marriage in Bangladesh [41], which may further elevate STI infections. Therefore, providing financial support and promoting access to education, especially for poorer families with teenage girls, can significantly reduce STIs’ excess burden among these vulnerable population groups.

Our analyses also revealed that richer women were more advantageous in terms of STI risk than the poorest women. This finding is supported by the evidence generated from other studies [18,27]. One plausible explanation is that higher wealth, which facilitates access to better education, increased access to mass media, and improved healthcare facilities, may contribute to increasing women’s awareness about STI and seeking better STI treatment [29]. Another plausible explanation is that wealthier women mainly live in cities and may experience less societal and cultural pressure to adapt to traditional gender norms than poorer women who primarily live in rural areas [9]. This advantage in living in cities may enhance the wealthiest women’s capacity to practice safe sex. For this reason, we recommend specific awareness and education campaigns targeting the poorest women, specifically for those who live in rural areas, to reduce the likelihood of STI symptoms. Including more information about STIs in the national curriculum and community-based STI education campaigns through mass media can be two potential ways to reach them across the country and provide adequate STI information for improving their STI knowledge. 

Importantly, our study found that women who used traditional (e.g., abstinence) and modern (e.g., condoms and oral pills) contraceptives reported higher abnormal genital discharge than those who used no contraceptives. These findings agree with a previous study [18]. Women who use contraceptives tend to be more aware of their sexual and reproductive health complications. As such, they are more likely to report STI symptoms [42]. This can be explained by the fact that women’s use of oral contraceptive pills [43] and inconsistent condom use, number of partners, and limited access to STI testing by their husbands [37,44] may increase STI risks among the former. It can be noted that women’s STI infections may also prompt their husbands to use condoms and practice safe intercourse [18]. Therefore, we recommend specific interventions for women who use traditional and modern contraceptive methods. 

Our study’s finding also demonstrates that the couples’ joint decision-making regarding women’s healthcare is associated with decreased STI symptoms. Previous research did not explicitly tease out couples’ joint decisions regarding women’s healthcare and its association with STI symptoms. Evidence indicates that joint decision-making can facilitate women’ access to STI care and testing [45] by increasing their health-seeking behaviors [46]. When decisions are jointly made with husbands/partners, women reported a lower rate of STI symptoms, underscoring the importance of gender equality in marital and sexual relationships [17] and women’s sole decision-making dynamics in their healthcare [46]. Respecting women’s preferences and healthcare decisions by both members of a couple can also produce better reproductive health outcomes [46,47]. Our study’s finding has policy implications because women’s preferences sole and joint decision-making (with husbands) regarding their healthcare may reduce STI infection and improve their sexual and reproductive health. 

Women’s STIs cannot be considered in isolation from men’s STIs [48]. In patriarchal settings such as Bangladesh, men tend to transmit STIs to their female partners [49]. The proportion of having premarital and unsafe sex tends to be higher among men than women in Bangladesh [37]. However, there is a striking lack of data about men’s STI infection, highlighting the additional need to collect and research men’s STI infection. The Bangladesh Bureau of Statistics (BBS) needs to compile information about men’s health to allow cross-sex comparisons of STIs. 

Similar to previous studies [9,27], our study substantiated that women who reported that wife-beating was justified if they refused to have sex had a higher likelihood of having STI symptoms such as genital sores/ulcers. This may be due to women’s financial dependence on men, patriarchal norms, and limited negotiation skills, thus limiting women’s ability to refuse intercourse against their will [9,50]. This is supported by the existing evidence that indicates that abused women are more likely to have unprotected intercourse, which may heighten their STI risks [50]. Disseminating information for creating awareness among men regarding the adverse consequences of violence against women via mass media (e.g., televisions, newspapers, and radios), social media (e.g., Facebook and Twitter), and leaflets and posters might be a potential option for changing men’s attitudes toward women and minimizing the risk of violence against women.

### Strengths and Limitations

This study has several strengths and limitations. The key strength is that used the three most recent rounds of the BDHS data (2007, 2011, and 2014), representative of the general population. Therefore, the findings are generalizable to the women of reproductive age in Bangladesh and similar settings. Furthermore, we were able to adjust models for a range of potential covariates. However, we acknowledge that our findings and interpretations should be considered in light of several notable limitations. First, STI symptoms and women’s demographic, socioeconomic, and behavioral characteristics were self-reported, often subject to recall bias and nondisclosure. Self-reporting has possibly resulted in social desirability bias or underestimation of the prevalence as STIs remain a taboo subject in Bangladesh. Second, although there are many STI symptoms, we considered only two symptoms, including abnormal genital discharge and genital sores/ulcers. Moreover, our analysis is based on STI symptoms, and sometimes STI symptoms remain undiagnosed or untreated. This may make the analysis somewhat weak. Third, this study extracted data from the three independent rounds of BDHS (2007, 2011, and 2014), and some households may have been included in all three rounds of the BDHS. This might cause an overestimation of our outcomes of interest, especially if there was significant overlapping of the sampling unit. Fourth, we do not know the extent of abnormal genital discharge and genital sores/ulcers of the participants from the current study, making the findings weak. Fifth, we cannot establish the causal mechanisms through which the risk factors influence STI and the self-reported STI symptoms due to the nature of cross-sectional data. These limitations highlight the need for further research by using longitudinal data to establish causal relationships between several STI symptoms in women and socioeconomic and behavioral risk factors. Future research should also be focused on measuring the prevalence of STI symptoms through clinical evidence. Thus, future research may provide a better understanding of the prevalence of STI symptoms and their associated risk factors and further public health efforts.

## 5. Conclusions

This study advances previous work by providing the prevalence of STI symptoms in ever-married women in Bangladesh and substantial evidence about the association of STI symptoms with various demographic, socioeconomic, and behavioral risk factors. The study found that one in ten ever-married Bangladeshi women had STI symptoms in the past twelve months prior to the survey. Women from the middle-aged group, with lower socioeconomic status, married at an earlier age, and having decision-making capacity about own healthcare had a heightened likelihood of STI symptoms. 

Our findings have important implications for clinical and public health interventions to improve sexual and reproductive health among vulnerable population groups. There is an urgent need to launch clinical and social–cultural intervention programs for these Bangladeshi women of reproductive age. Proper diagnosis and treatment for STIs can also help reduce the STI symptoms in women. However, the diagnosis and treatment programs reducing STI will not succeed if they do not consider the targeted SRH services of women of reproductive age and facilitate their access to these services. Additionally, to address this public health problem, attempts can be made to educate and empower women through community outreach programs that discuss the harmful effects of child marriage on women’s STI symptoms. These programs may also discuss how couples’ joint decisions regarding healthcare may help reduce the risk of having STI symptoms in women.

## Figures and Tables

**Figure 1 ijerph-19-01906-f001:**
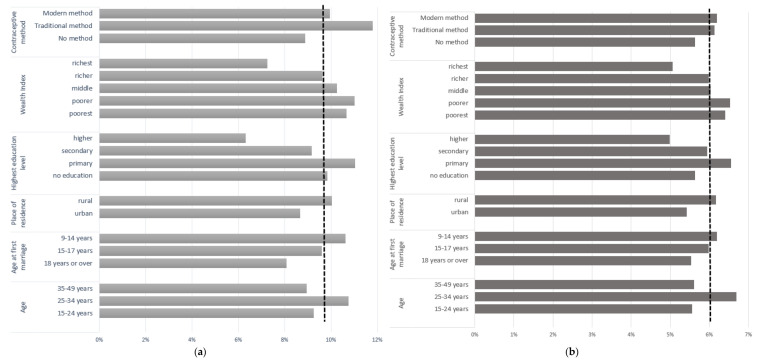
Prevalence of STI symptoms among ever-married women aged 15–49 years in Bangladesh during 2007–2014 by key risk factors: (**a**) abnormal genital discharge; (**b**) genital sores/ulcers.

**Table 1 ijerph-19-01906-t001:** A list of explanatory variables.

	Variables	Measuement Scales	Descriptions
Demographic factors	Age	Categorical	Three age groups are recoded as 15–24, 25–34, 35–49 years, with 15–24 years as a reference group.
	Age at first marriage	Categorical	Age at first marriage is grouped as 9–14, 15–17, 18 years or older, with 18 years or older as reference group.
	Type of residence	Binary	Type of residence is recoded into a dummy variable (rural = 0 and urban = 1), with rural as a reference group.
Socioeconomic factors	Education	Categorical	Education has four groups: no education (reference), primary, secondary, higher.
	Partner’s education	Categorical	Partner’s education is recoded as no education (reference), primary, secondary, higher.
	Wealth quintile	Categorical	Wealth quintile is categorized into five groups: poorest (reference), poorer, middle, richer, richest.
	Paidwork status	Binary	Paidwork is recoded into a dummy variable: no = 0 (reference), yes = 1.
Behavioral factors	Contraceptive method	Categorical	Contraceptive method has three categories such as no method (reference), traditional method, modern method.
	Knowledge about STI	Binary	Knowledge about STI is coded as a binary variable, with no = 0 (reference), yes = 1.
	Wife beating justified	Binary	Wife beating justified has two groups: with no = 0 (reference), yes = 1.
	Women’s healthcare decision-making	Categorical	Women’s healthcare decision-making is categorized into 4 groups: wife (reference), wife and husband, respondent and someone else, and husband.
	Exposure to mass media	Categorical	Exposure to mass media is coded as not at all (reference), irregular, regular.
	Survey year	Categorical	Survey year has three groups: 2007 (reference), 2011, 2014.

**Table 2 ijerph-19-01906-t002:** Characteristics of the survey participants by STI symptoms.

	Had Abnormal Genital Discharge in the Past 12 Months Prior to the Survey (n = 41,777)			Had Genital Sores/Ulcers in Past 12 Months Prior to the Survey (n = 41,777)		
	No (*n* = 37,598)	Yes (*n* = 4179)	*p*-Value	No (*n* = 39,155)	Yes (*n* = 2622)	*p*-Value
Demographic factors						
Age			<0.001			<0.001
15–24 years	11,981 (32.8%)	1306 (31.2%)		12,525 (32.8%)	762 (31.2%)	
25–34 years	13,070 (34.6%)	1606 (38.9%)		13,642 (34.6%)	1034 (38.9%)	
35–49 years	12,547 (32.6%)	1267 (29.9%)		12,988 (32.6%)	826 (29.9%)	
Age at first marriage						
18 years and older	8726 (21.5%)	803 (17.6%)		8986 (21.5%)	543 (17.6%)	
15–17 years	14,804 (39.1%)	1652 (38.8%)		15,422 (39.1%)	1034 (38.8%)	
9–14 years	14,068 (39.4%)	1724 (43.6%)		14,747 (39.4%)	1045 (43.6%)	
Type of residence			<0.001			<0.001
urban	13,581 (26.4%)	1341 (23.4%)		14,072 (26.4%)	850 (23.4%)	
rural	24,017 (73.6%)	2838 (76.6%)		25,083 (73.6%)	1772 (76.6%)	
Socioeconomic factors						
Education			<0.001			<0.001
No education	9091 (26.3%)	1038 (26.8%)		9516 (26.3%)	613 (26.8%)	
Primary	10,997 (29.3%)	1388 (33.9%)		11,514 (29.3%)	871 (33.9%)	
Secondary	13,860 (36.3%)	1522 (34.2%)		14,433 (36.3%)	949 (34.2%)	
Higher	3650 (8.2%)	231 (5.1%)		3692 (8.2%)	189 (5.1%)	
Partner’s education			<0.001			<0.001
No education	10,438 (29.7%)	1243 (31.8%)		10,939 (29.7%)	742 (31.8%)	
Primary	10,113 (27.0%)	1302 (30.8%)		10,633 (27.0%)	782 (30.8%)	
Secondary	11,032 (28.9%)	1191 (27.5%)		11,457 (28.9%)	766 (27.5%)	
Higher	6015 (14.3%)	443 (9.9%)		6126 (14.3%)	332 (9.9%)	
Wealth quintile			<0.001			<0.001
Poorest	6106 (17.4%)	788 (19.4%)		6423 (17.4%)	471 (19.4%)	
Poorer	6856 (19.0%)	883 (21.9%)		7193 (19.0%)	546 (21.9%)	
Middle	7353 (20.1%)	866 (21.4%)		7689 (20.1%)	530 (21.4%)	
Richer	7915 (21.1%)	911 (21.0%)		8272 (21.1%)	554 (21.0%)	
Richest	9368 (22.4%)	731 (16.3%)		9578 (22.4%)	521 (16.3%)	
Paid work status			<0.001			<0.001
No	30,244 (79.0%)	3280 (76.5%)		31,490 (79.0%)	2034 (76.5%)	
Yes	7354 (21.0%)	899 (23.5%)		7665 (21.0%)	588 (23.5%)	
Behavioral factors						
Contraceptive method			<0.001			<0.001
No method	15,020 (40.3%)	1539 (36.7%)		15,603 (40.3%)	956 (36.7%)	
Traditional method	3203 (8.5%)	419 (10.6%)		3387 (8.5%)	235 (10.6%)	
Modern method	19,375 (51.2%)	2221 (52.7%)		20,165 (51.2%)	1431 (52.7%)	
Knowledge about STI			<0.001			<0.001
No	10,254 (28.6%)	1160 (29.2%)		10,759 (28.6%)	655 (29.2%)	
Yes	27,344 (71.4%)	3019 (70.8%)		28,396 (71.4%)	1967 (70.8%)	
Wife beating justified if she refuses to have sex			<0.001	0 (0.0%)		<0.001
No	34,526 (92.0%)	3805 (91.4%)		35,967 (92.0%)	2364 (91.4%)	
Yes	3072 (8.0%)	374 (8.6%)		3188 (8.0%)	258 (8.6%)	
Decisions regarding own health care			<0.001			<0.001
Wife	5273 (13.4%)	676 (15.6%)		5523 (13.4%)	426 (15.6%)	
Wife and husband	18,142 (48.2%)	1797 (44.2%)		18,748 (48.2%)	1191 (44.2%)	
Respondent and someone else	2844 (8.1%)	286 (7.0%)		2973 (8.1%)	157 (7.0%)	
Husband	11,339 (30.3%)	1420 (33.2%)		11,911 (30.3%)	848 (33.2%)	
Exposure to mass media			<0.001			<0.001
Not at all	12,879 (35.1%)	1485 (36.0%)		13,452 (35.1%)	912 (36.0%)	
Irregular	3892 (10.4%)	536 (12.7%)		4127 (10.4%)	301 (12.7%)	
Regular	20,827 (54.4%)	2158 (51.3%)		21,576 (54.4%)	1409 (51.3%)	

Note: *p*-value for chi-square test.

**Table 3 ijerph-19-01906-t003:** Multivariable logistic regression models assessing associations of abnormal genital discharge with demographic, socioeconomic, and behavioral risk factors.

	Model 1	95% CI	Model 2	95% CI	Model 3	95% CI
Demographic factors						
Age (Ref: 15–24 years)						
25–34 years	1.17 **	(1.06–1.30)	1.17 **	(1.05–1.29)	1.14 **	(1.03–1.27)
35–49 years	0.94	(0.85–1.03)	0.94	(0.84–1.04)	0.91	(0.82–1.02)
Age at first marriage (Ref: 18 years or older)						
9–14 years	1.41 ***	(1.27–1.56)	1.22 ***	(1.09–1.37)	1.21 ***	(1.08–1.36)
15–17 years	1.23 ***	(1.09–1.38)	1.11	(0.99–1.25)	1.11	(0.98–1.24)
Type of residence (Ref: Rural)						
Urban	1.16 **	(1.04–1.28)	1.01	(0.90–1.13)	1.02	(0.91–1.15)
Socioeconomic factors						
Education (Ref: No education) ^a^						
Primary			1.12	(0.99–1.26)	1.08	(0.95–1.23)
Secondary			1	(0.88–1.14)	0.95	(0.83–1.09)
Higher			0.84	(0.67–1.05)	0.79*	(0.63–1.00)
Partner’s education (Ref: No education) ^a^						
Primary			1.07	(0.97–1.19)	1.06	(0.96–1.18)
Secondary			0.97	(0.86–1.10)	0.96	(0.85–1.08)
Higher			0.86	(0.72–1.03)	0.84	(0.71–1.01)
Wealth quintile (Ref: Poorest)						
Poorer			1.06	(0.94–1.20)	1.04	(0.92–1.18)
Middle			1.01	(0.88–1.15)	0.95	(0.83–1.10)
Richer			0.97	(0.85–1.12)	0.9	(0.78–1.04)
Richest			0.79 **	(0.67–0.94)	0.74 ***	(0.62–0.88)
Paid work status (Ref: No) ^c^						
Yes			1.15 **	(1.04–1.28)	1.13 *	(1.02–1.26)
Behavioral factors						
Contraceptive method (Ref: No method)						
Traditional method					1.42 ***	(1.23–1.64)
Modern method					1.09 *	(1.00–1.19)
Knowledge about STI (Ref: No) ^a^						
Yes					1.1	(0.98–1.22)
Wife beating justified (Ref: No) ^b^						
Yes					1.03	(0.90–1.18)
Women’s healthcare decision-making (Ref: Wife) ^a^						
Wife and husband					0.76 ***	(0.68–0.85)
Respondent and someone else					0.85	(0.71–1.02)
Husband					0.91	(0.80–1.04)
Exposure to mass media (Ref: Not at all)						
Irregular					1.22 **	(1.07–1.38)
Regular					1.11 *	(1.00–1.23)
Survey year (Ref: 2007)						
Survey Year = 2011	1.47 ***	(1.29–1.66)	1.49 ***	(1.31–1.69)	1.49 ***	(1.31–1.69)
Survey Year = 2014	1.57 ***	(1.37–1.79)	1.54 ***	(1.35–1.76)	1.56 ***	(1.37–1.78)
Constant	0.05 ***	(0.05–0.06)	0.07 ***	(0.05–0.08)	0.07 ***	(0.05–0.09)
Observations	41,777		41,777		41,777	

Note: *** *p* < 0.001, ** *p* < 0.01, * *p* < 0.05; ^a^ less than 0.1% missing values; ^b^ less than 1% missing values; ^c^ less than 4% missing values.

**Table 4 ijerph-19-01906-t004:** Multivariable logistic regression models assessing associations of genital sores/ulcers with demographic, socioeconomic, and behavioral risk factors.

	Model 1	95% CI	Model 2	95% CI	Model 3	95% CI
Demographic factors						
Age (Ref: 15–24 years)						
25–34 years	1.22 ***	(1.09–1.37)	1.24 ***	(1.10–1.40)	1.21 **	(1.06–1.37)
35–49 years	1.00	(0.89–1.13)	1.06	(0.93–1.22)	1.05	(0.91–1.21)
Age at first marriage (Ref: 18 years or older)						
9–14 years	1.13	(0.99–1.28)	1.07	(0.94–1.23)	1.06	(0.93–1.22)
15–17 years	1.09	(0.96–1.24)	1.05	(0.92–1.19)	1.04	(0.92–1.19)
Type of residence (Ref: Rural)						
Urban	1.14 *	(1.01–1.28)	1.05	(0.92–1.20)	1.08	(0.95–1.23)
Socioeconomic factors						
Education (Ref: no education) ^a^						
Primary			1.19 *	(1.03–1.37)	1.14	(0.99–1.31)
Secondary			1.17	(0.99–1.39)	1.09	(0.92–1.28)
Higher			1.09	(0.83–1.42)	1.01	(0.77–1.32)
Partner’s education (Ref: No education) ^a^						
Primary			1.07	(0.94–1.22)	1.06	(0.93–1.21)
Secondary			1.02	(0.88–1.19)	1.01	(0.86–1.17)
Higher			0.95	(0.77–1.19)	0.94	(0.76–1.17)
Wealth quintile (Ref: Poorest)						
Poorer			1.01	(0.87–1.18)	0.99	(0.85–1.15)
Middle			0.93	(0.79–1.09)	0.88	(0.75–1.04)
Richer			0.93	(0.78–1.10)	0.86	(0.72–1.04)
Richest			0.81 *	(0.67–0.99)	0.75 **	(0.61–0.93)
Paid work status (Ref: No) ^c^						
Yes			1.23 **	(1.09–1.38)	1.20 **	(1.07–1.36)
Behavioral factors						
Contraceptive method (Ref: No method)						
Traditional method					1.09	(0.92–1.31)
Modern method					1.07	(0.96–1.18)
Knowledge about STI (Ref: No) ^a^						
Yes					1.26 ***	(1.11–1.42)
Wife beating justified (Ref: No) ^b^						
Yes					1.27 **	(1.08–1.49)
Women’s healthcare decision-making (Ref: wife) ^a^						
Wife and husband					0.81 **	(0.70–0.93)
Respondent and someone else					0.78 *	(0.62–0.98)
Husband					0.88	(0.76–1.02)
Exposure to mass media (Ref: Not at all)						
Irregular					1.02	(0.86–1.20)
Regular					1.05	(0.92–1.20)
Survey year (Ref: 2007)						
Survey Year = 2011	1.35 ***	(1.16–1.56)	1.37 ***	(1.18–1.59)	1.38 ***	(1.19–1.59)
Survey Year = 2014	1.05	(0.90–1.23)	1.02	(0.88–1.19)	1.03	(0.88–1.20)
Constant	0.04 ***	(0.04–0.05)	0.04 ***	(0.03–0.05)	0.04 ***	(0.03–0.06)
Observations	41,777		41,777		41,777	

Note: *** *p* < 0.001, ** *p* < 0.01, * *p* < 0.05; ^a^ less than 0.1% missing values; ^b^ less than 1% missing values; ^c^ less than 4% missing values.

## Data Availability

The datasets used and analyzed in this study are available on the MEASURE DHS website: https://dhsprogram.com/data/available-datasets.cfm (accessed on 30 January 2022).

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
