# Peer review of "Prevalence and Demographic, Socioeconomic, and Behavioral Risk Factors of Self-Reported Symptoms of Sexually Transmitted Infections (STIs) among Ever-Married Women: Evidence from Nationally Representative Surveys in Bangladesh"

_ijerph, 2022, doi:10.3390/ijerph19031906_

Round 1

Reviewer 1 Report

Thank you for the opportunity to review your work. I think it has merit and can be improved with some minor revisions to language and content.

1. There are a few typesetting errors in the authors' names (Dr Hasan's first name is not abbreviated with a period, there is a floating "and" after the names).
2. Repeated reference to "STI symptoms" is unclear. Are you worried about the symptoms or the fact that they indicate there is an STI? For example, do you truly mean in the first sentence of the abstract that STI symptoms are a major public health concern, or that STIs are a major public health concern. And if it is the former, why? Examples of symptoms should be given earlier. It sounds like you are using symptoms as a proxy for infection, since sometimes infections go undiagnosed or untreated.
3. There are small but noticeable errors in English grammar that will need correction (eg, "genital sore/ulcer" in the abstract needs either an indefinite article or plural noun). Note that that particular example is corrected in section 2.2, so I would recommend a review of all terms, as it is back to being poorly phrased in section 3.1.
4. It is debated whether BV is truly an STI. It is also unnecessary to give gonorrhea as an example of an STI twice in the first paragraph.
5. I really appreciate that you point out the limitations of previous studies (final paragraph of section 1). Since yours may be the first of its kind, how did you examine the literature to make sure such work has not been published ("to the best of [your] knowledge")?
6. Do you know to what level "genital discharge" and "genital sores/ulcers" were described to survey respondents? Is it possible it was misunderstood that "discharge" was pain, and thereby making the survey flawed?
7. It might be more clear to list the variables in section 2.3 in a table or box.
8. Please advise the reader that Stata is statistical software.
9. Please adjust the carriage return for the first "yes" column (and various factors on the far left) in Table 1. Consider widening the table to make the demographics easier to read.
10. Abbreviations like CI and AOR do not appear to be defined earlier. Again, just make that clear for the reader.
11. Please consistently capitalize "Model" for Models 1, 2, and 3 (section 3.3).
12. Please either capitalize all of COVID-19, or refer to it as SARS-CoV-2 (section 4).
13. Would you recommend any campaigns targeting men, specifically those who think violence towards their wife is permissible?

Overall, I think this is a nice paper that has promise. What is needed most is a review of grammar for clarity and consistency, followed by further discussion as to generalizability of the results and/or next steps in research and public health efforts.

Author Response

3 February 2022

Editor-in-Chief

Dear Editor/reviewers,

We thank you for the opportunity to revise and resubmit the manuscript. We greatly appreciate the valuable comments from the reviewers. We have made edits for typos, corrected grammar for clarity and modified the enclosed manuscript accordingly (all with track changes). In the following table, we include reviewers’ comments and our responses to their comments.

We are confident that the revisions have improved the quality of our manuscript. We hope that our revisions have adequately addressed reviewers’ comments/concerns. We look forward to hearing from you with a final decision regarding the acceptance of our manuscript.

Regards,

Authors

Reviewer 2 Report

The manuscript reports about “Prevalence and demographic, socio-economic, and behavioral risk factors of [self-reported!] symptoms of sexually transmitted infections (STIs) among ever-married women” based on three public samples (2007, 2022, 2014) from the DHS program for Bangladesh that were combined (https://dhsprogram.com/data/available-datasets.cfm).

While self-reported STI symptoms and related factors are a relevant topic for IJERPH, the manuscript, unfortunately, comes with several problems.

  1. The manuscript does not provide any definitions of core concepts such as “STI”, “STI diagnosis”, “self-reported STI diagnosis”, and “self-reported STI symptoms” and their relations. Authors only talk about “STI symptoms” without any explanation and contextualization of this concept versus related concepts.
  2. It should be transparent in the title that the study is about *self-reported* STI symptoms.
  3. The manuscript does not clearly explain the validity of self-reported “STI symptoms” such as vaginal sore/ulcer and vaginal discharge. Vaginal discharge, for example, is not necessarily a symptom of an STI but can be normal during the menstrual cycle. Issues of validity of main outcomes must be explained in greater detail.
  4. The referenced “global DHS program” needs more background information and an online link.
  5. It is not very convincing to merge the samples of three waves of the same survey as there is participant overlap and significant time delay. Better use the most current sample from 2014 only. Use the data sets from 2011 and 2007 to replicate the model.
  6. One main problem is the lack of theoretical background. The selection of variables is not clearly explained and remains often unclear why and via which causal mechanism they should influence STI contraction and hence self-reported STI symptoms.
  7. Mostly post hoc speculations are provided.
  8. It remains unclear why “healthcare decision-making autonomy” and “joint decision-making” are labeled as such. If women don’t have decision making autonomy this is a violation of their human rights and not “joint decision making”. Also the data reported in Table 2 and the claim in the abstract seem to be contradictory.
  9. Layout of tables should be improved.
  10. No cultural background is provided in the introduction. In the post-hoc speculations – all of a sudden – authors talk about Bangladesh as a “patriarchal setting” without explanation.

Author Response

(The authors gave the same response as above.)

Round 2

Reviewer 2 Report

Thanks for the answer letter that clarified most of my concerns. The revision has sufficiently improved the manuscript.